# How do and could clinical guidelines support patient-centred care for women: Content analysis of guidelines

**Anna R. Gagliardi**[1]*, **Courtney Green**[2], **Sheila Dunn**[3], **Sherry L. Grace**[4], **Nazilla Khanlou**[4], **Donna E. Stewart**[1]

**1** University Health Network and University of Toronto, Toronto, Ontario, Canada, **2** Society of Obstetricians and Gynaecologists of Canada, Ottawa, Ontario, Canada, **3** Women's College Hospital, Toronto, Ontario, Canada, **4** York University and University Health Network, Toronto, Ontario, Canada

☯ These authors contributed equally to this work.
* anna.gagliardi@uhnresearch.ca

**Data Availability Statement:** All relevant data are within the manuscript and its Supporting Information files.

## Abstract

### Objectives

Patient-centred care (PCC) improves multiple patient and health system outcomes. However, many patients do not experience PCC, particularly women, who are faced with disparities in care and outcomes globally. The purpose of this study was to identify if and how guidelines address PCC for women (PCCW).

### Methods

We searched MEDLINE, EMBASE, National Guideline Clearing House, and guideline developer websites for publicly-available, English-language guidelines on depression and cardiac rehabilitation, conditions with known gendered inequities. We used summary statistics to report guideline characteristics, clinical topic, mention of PCC according to McCormack's framework, and mention of women's health considerations. We appraised guideline quality with the AGREE II instrument.

### Results

A total of 27 guidelines (18 depression, 9 cardiac rehabilitation) were included. All 27 guidelines mentioned at least one PCC domain (median 3, range 1 to 6), most frequently exchanging information (20, 74.1%), making decisions (20, 74.1%), and enabling patient self-management (21, 77.8%). No guidelines fully addressed PCC: 9 (50.0%) of 18 depression guidelines and 3 (33.3%) of 9 cardiac rehabilitation guidelines addressed 4 or more PCC domains. Even when addressed, guidance was minimal and vague. Among 14 (51.9%) guidelines that mentioned women's health, most referred to social determinants of health; none offered guidance on how to support women impacted by these factors, engage women, or tailor care for women. These findings pertained even to women-specific guidelines. Reported use or type of guideline development process/system did not appear to be linked with PCCW content. Based on quality appraisal with AGREE II, guidelines were either

**Funding:** This work received funding from the Ontario Ministry of Health & Long Term Care (grant 251) to ARG. The funder had no role in study design, data collection and analysis, decision to publish, or preparation of the manuscript.

not recommended or recommended with modifications. In particular, the stakeholder involvement AGREE II domain was least addressed, but guidelines that scored higher for stakeholder involvement also appeared to better address PCCW.

## Implications

This research identified opportunities to generate guidelines that achieve PCCW. Strategies include employing a PCC framework, considering gender issues, engaging women on guideline-writing panels, and including patient-oriented tools in guidelines. Primary research is needed to establish what constitutes PCCW.

## Introduction

Patient-centred care (PCC) has been defined as healthcare that establishes a partnership among practitioners, patients and their families to ensure that care is attentive to the needs, values and preferences of patients [1,2]. PCC is considered a key element of high quality health care because it has been associated with beneficial patient (knowledge, relationship with providers, service experience and satisfaction, treatment compliance, health outcomes) and health system (cost-effective service delivery) outcomes [3–5]. PCC is characterized by the patient-provider relationship (sharing information, empathy, empowerment), partnership (sensitivity to needs, patient involvement in care), and health promotion (case management, patient empowerment) [6]. McCormack et al. used rigorous methods to establish a comprehensive PCC framework that included 31 sub-domains within six interdependent domains: fostering healing relationships, exchanging information, recognizing and responding to patient emotions, managing uncertainty, making decisions, and enabling patient self-management [7].

Despite the benefits associated with PCC and insight on the components of PCC, research shows that many patients do not receive PCC. A national survey in the United States showed that, among 2,718 responding adults aged 40 or greater with 10 common medical conditions, there was considerable variation in perceived PCC among patients including involvement in discussing treatment options and making decisions, and women were less likely to experience PCC [8]. The 2009 World Health Organization report, *Women and Health* emphasized a need to improve the quality of women's health care services and women's health [9]. For example, over-medicalization of women-specific conditions has led to the creation and overtreatment of so-called "diseases" (i.e. menopause), and confusion and anxiety among women about how to maximize their health [10]. For other conditions common to men and women such as cardiovascular disease, women are less often referred for diagnostic and therapeutic interventions [11]. Monitoring by the United Nations continues to show that gender-imposed disparities influence women's health. As a result, improving care for women remains a priority in their 2018 report, *Gender Equality in the 2030 Agenda for Sustainable Development* [12].

PCC for women (PCCW) stands to improve women's health care experience and outcomes; thus, strategies are needed to promote and support PCCW. The *Ontario Women's Health Framework* issued four recommendations on how to achieve PCCW: consider gender and health in all government policies; adopt quality measures that reflect women's priorities; share information with women directly; and develop and implement clinical guidelines that include specific evidence-based gender elements [13]. Clinical guidelines are defined as systematically developed statements to assist practitioner and patient decisions about appropriate health care for specific clinical circumstances [14]. Guidelines have been referred to as one of the

foundations for efforts to improve health care because they synthesize scientific evidence and offer recommendations that serve as the basis not only for supporting patient-clinician decision-making, but also for planning, evaluating and improving health care quality [15].

However, we [16] and countless others [17–19] have evaluated the quality of guidelines on numerous clinical topics, and found that many aspects of guidelines could be improved, particularly stakeholder involvement, which refers to incorporating the views of end-users including patients so that guidelines are more patient-centred [16]. To date, no study has assessed whether and how guidelines, fundamental tools for optimizing patient care and associated outcomes, address PCCW. The purpose of this study was to analyze guidelines for content pertaining to PCC and/or women's health and assess the quality of those guidelines including stakeholder involvement. If guidance for PCC and/or women's health is absent, this may reveal opportunities for developers to enhance their guidelines with content that supports PCCW.

## Methods

### Approach

We employed a qualitative content analysis approach, which is a method of studying written or visual communication, to describe whether and how Canadian and international guidelines address women's health, gendered inequities, or PCCW [19]. This approach involved screening, data extraction and data analysis of guidelines [20]. While not a typical synthesis, the Preferred Reporting Items for Systematic Reviews and Meta-Analyses (PRISMA) criteria guided the conduct and reporting of the methods and results [21]. Data were publicly available so institutional review board approval was not necessary.

### Eligibility criteria

We included guidelines published in English language after January 1, 2010. This date was chosen because recommendations in our jurisdiction [13] and internationally [9] published in 2009 advocated for guidelines to consider issues of gender and health. We included guidelines on two conditions that affect men and women across the lifespan: depression (often present in the post-natal period among women) and cardiovascular disease including rehabilitation (now affecting women in middle age). We chose these topics because they have been associated with known gendered inequities in quality of care in Canada and elsewhere: when women report depression, it is more likely to be dismissed as stress compared with men who are more likely to receive treatment; and compared with men, women are less likely to be referred to cardiac rehabilitation [22–24]. These topics were also recommended by the research team, which included researchers with expertise in the conditions of interest, Chairs of Women's Health, and representatives of quality improvement and professional organizations. Eligible guidelines were developed by non-profit organizations including government, professional societies, disease-specific foundations or quality improvement/monitoring agencies in Canada and in English-language countries with comparative health care contexts including Australia, New Zealand, England, Scotland, Ireland and United States. Guidelines were not eligible if they were only available in languages other than English, were not publically available, and were specific to topics that were not the conditions of interest, were conducted in low-resource countries or other non-comparative health care contexts, or were based on consensus-only.

### Searching and screening

Guidelines were identified by a research assistant (MZ) using two strategies: searching indexed databases and searching a repository of international guidelines. MEDLINE and EMBASE

were searched on June 12, 2018 for guidelines published from January 1, 2010 to that date. We also identified guidelines in the National Guideline Clearinghouse, a comprehensive database of international guidelines (https://www.ahrq.gov/gam/index.html was terminated on July 16, 2018) using both searches and browsing of disease-specific lists, and then following links to developer web sites to acquire guidelines. Searches were executed in June 2018. All search strategies are included in S1 Table. MZ captured the results of all searches in an Excel file, and titles and/or abstracts were independently screened by MZ and ARG.

## Data extraction

A data extraction form was developed to collect information on guideline characteristics (year of publication, country, clinical topic, development process/system). Data on PCC and women's health were extracted using a summative qualitative approach, meaning that the text pertaining to these concepts was extracted [20]. As a pilot test, MZ, JR, BN, DK (also research assistants) and ARG independently extracted data from five guidelines, compared and discussed results to establish a shared understanding of what to extract. Guidelines were perused for any mention of PCC according to the McCormack et al. framework of 31 elements organized in the six domains, chosen because it was rigorously developed [7] and more comprehensive than other PCC frameworks [6,25,26] (S2 Table). Guidelines were also perused for any mention of women's health including issues to consider when delivering, overseeing or supporting the care of women; gender issues related to education, socioeconomic status, ethnicity or literacy; frameworks or models of women's health or PCCW; or any guidance on how to engage women or deliver care to women. MZ extracted data on PCC and women's health, which was independently checked by JR, and ARG resolved discrepancies.

## Quality assessment

The methodological quality of included guidelines was assessed with AGREE II (Appraisal of Guidelines for Research and Evaluation), a rigorously-developed and widely used instrument comprised of six domains: scope and purpose, stakeholder involvement, rigour of development, clarity of presentation, applicability, and editorial independence [14]. In particular, we were interested in stakeholder involvement, which leads to guidelines that are more patient-centred. Two individuals (DK and MZ) independently appraised guidelines based on instructions in the AGREE II instruction manual [14]. As a pilot test, BN, DK and MZ first independently assessed 10 guidelines, then compared and discussed their findings to achieve a shared understanding of how to apply the criteria and resolve discrepancies. DK and MZ then assessed the remaining guidelines for 23 items across six domains via a seven-point Likert scale from strongly disagree (1) to strongly agree (7) that the item was met. ARG resolved discrepancies. Based on these scores, an overall rating of quality was given to each guideline, and a recommendation whether to use, use with modifications, or not use each guideline. Individual ratings across all 23 items for each guideline were combined to yield an overall average appraisal score. To determine scaled domain percentages, both appraisers' ratings of items within each domain were summed, and the maximum and minimum possible domain scores were scaled before converting this into an overall percentage for the domain.

## Data analysis

Summary statistics were used to report guideline publication date, country, condition, and development process/system. Content related to PCC domains and women's health were reported using summary statistics for guidelines overall, by condition, and development process/system. To assess if quality appraisal results (stakeholder involvement in particular) were

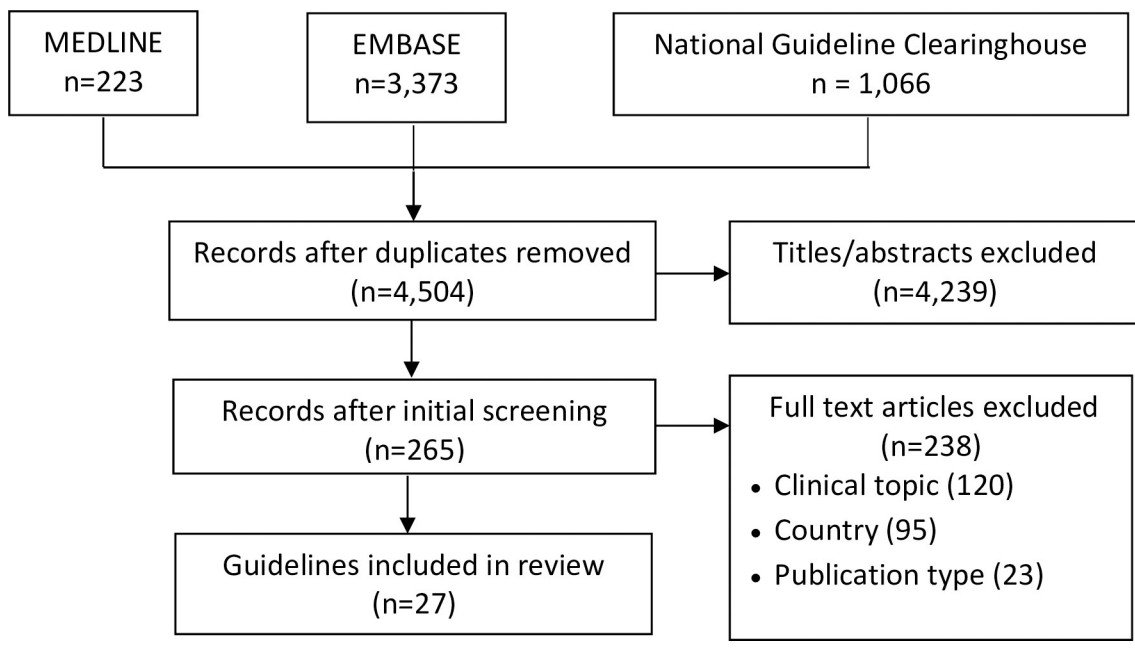

**Fig 1. PRISMA diagram.**

related to inclusion of PCC and women's health, these were cross-compared and reported using summary statistics.

## Results

### Search results

Searching resulted in 4,662 guidelines, of which 4,504 were unique, and 4,239 were excluded by title screening. Among 265 full-text guidelines, 238 were excluded because the clinical topic (120), country (95) or publication type (23) were not eligible (Fig 1). A total of 27 guidelines were eligible for review [27–54]. Extracted data are included in S3 Table.

### Guidelines characteristics

Guidelines were published between 2010 and 2017 (Table 1). Clinical topics included depression (18, 66.7%) and cardiac rehabilitation (9, 33.3%). Most of the guidelines were from Canada (8, 29.6%) and the United States (8, 29.6%).

**Table 1. Characteristics of included guidelines.**

| Characteristic | Clinical topic | | Total n (% of 27) |
|---|---|---|---|
| | **Mental Health (Depression) n (% of 18)** | **Cardiac Care (Rehabilitation) n (% of 9)** | |
| Canada | 8 (44.4) | -- | 8 (29.6) |
| United States | 5 (27.7) | 3 (33.3) | 8 (29.6) |
| England | 2 (11.1) | 1 (11.1) | 3 (11.1) |
| Scotland | 1 (2.7) | 2 (22.2) | 3 (11.1) |
| Australia and New Zealand | 2 (5.4) | -- | 2 (7.4) |
| Europe | -- | 2 (22.2) | 2 (7.4) |
| International group | -- | 1 (11.1) | 1 (3.7) |

## Patient-centred care

Table 2 summarizes whether and how PCC was addressed in included guidelines. All 27 guidelines mentioned at least one PCC domain (median 3, range 1 to 6). Three depression guidelines [31,33,44] and 1 cardiac rehabilitation guideline [48] mentioned all 6 PCC domains. Most guidelines considered the domains of exchanging information (20, 74.1%), making decisions (20, 74.1%), and enabling patient self-management (21, 77.8%). Fewer guidelines mentioned responding to emotions (14, 51.9%), and even fewer mentioned fostering a healing relationship (9, 33.3%) and managing uncertainty (7, 25.9%). A higher proportion of depression guidelines mentioned fostering a healing relationship, making decisions, and enabling patient self-management, while a higher proportion of cardiac rehabilitation guidelines mentioned exchanging information. Three of 18 depression guidelines focused on peri- or post-natal depression; and featured content for 2 or more PCC domains [28,31,42]. Two of 9 cardiovascular disease guidelines featured content for 1 and 3 PCC domains [45,51]. Thus, guidelines aimed at women did not apparently differ in PCC content from guidelines relevant to both women and men.

While all guidelines mentioned at least 1 aspect of PCC, it was not thoroughly addressed in any guidelines: 9 (50.0%) of 18 depression guidelines and 3 (33.3%) of 9 cardiac rehabilitation guidelines addressed 4 or more PCC domains. Even when PCC was addressed, guidance was often minimal and vague. For example, some guidelines emphasized that good communication between patients and clinicians is essential [33, 44], but provided no clear definition of what constitutes good communication, nor provided examples or instructions of how to initiate or facilitate communication. With regard to treatment decision-making, one guideline stated that "patient goals should be considered when choosing treatment" but did not include any additional information about how to engage patients in decision making [47].

**Depression.** Of 18 depression guidelines, 7 (38.9%) discussed aspects of fostering a healing relationship such as establishing a therapeutic alliance regardless of time constraints, and provide an open, non-judgmental environment for patients. Twelve (66.7%) guidelines mentioned exchanging information, recommending that information should be tailored to patient needs, culturally appropriate, and shared with families and care-givers when appropriate. Among 9 (50.0%) guidelines that mentioned responding to emotions, some stated that clinicians should provide support and encouragement, while others referred to destigmatizing depression by reassuring patients that it is not a personal weakness [33,36,38,44]. Managing uncertainty was mentioned in 5 (27.8%) guidelines; for example, clinicians should discuss the uncertainty of treatment effectiveness or prognosis [31,34,43,44] Fifteen (83.3%) guidelines discussed decision-making, with most emphasizing the importance of patient participation in treatment decision-making [27, 29, 31–33, 35, 36, 39, 40, 41, 44]. Among 15 (83.3%) guidelines that discussed enabling patient self-management, most recommended that clinicians should provide patients with follow-up plans and information to support self-management [31, 40, 44].

**Cardiac rehabilitation.** All 9 guidelines on cardiac management or rehabilitation included some mention of PCC. For facilitating a healing relationship, 2 (22.2%) guidelines stated that having a good relationship between patient and provider enables better communication, and is likely to influence the success of cardiac management [48,49]. Eight (88.9%) guidelines addressed the domain of exchanging information and most highlighted that good communication includes: listening to the patient, respecting views and beliefs, giving patients information they ask for or need in a way that they understand, confirming understanding via questions, defining unfamiliar words, writing down key works, and using diagrams [45–51]. Among 5 (55.6%) guidelines that mentioned responding to emotions, clinicians were

**Table 2. PCC and women's health content in included guidelines.**

| Guideline (year, country) | Patient-centred care domains (n,%) | | | | | | Total PCC domains (n) | Women's health |
|---|---|---|---|---|---|---|---|---|
| | Fostering relationship | Exchanging information | Addressing emotions | Managing uncertainty | Making decisions | Enabling self-management | | |
| **DEPRESSION** | | | | | | | | |
| Canadian Task Force on Preventive Health Care, 2013, Canada [27] | -- | ✓ | -- | -- | ✓ | ✓ | 3 | ✓ |
| BC Reproductive Mental Health Program & Perinatal Services BC, 2014, Canada [28] | -- | -- | ✓ | -- | -- | ✓ | 2 | ✓ |
| Toward Optimized Practice, 2015, Canada [29] | -- | -- | -- | -- | ✓ | ✓ | 2 | -- |
| BC Guidelines, 2013, Canada [30] | -- | ✓ | -- | -- | -- | ✓ | 2 | -- |
| The Centre of Perinatal Excellence, 2017, Australia [31] | ✓ | ✓ | ✓ | ✓ | ✓ | ✓ | 6 | ✓ |
| Canadian Partnership Against Cancer and the Canadian Association of Psychosocial Oncology, 2015, Canada [32] | -- | ✓ | ✓ | ✓ | -- | -- | 3 | -- |
| Royal Australian and New Zealand College of Psychiatrists, 2015, Australia & New Zealand [33] | ✓ | ✓ | ✓ | ✓ | ✓ | ✓ | 6 | ✓ |
| Registered Nurse's Association of Ontario, 2016, Canada [34] | ✓ | ✓ | ✓ | -- | ✓ | ✓ | 5 | -- |
| Canadian Network for Mood and Anxiety Treatments, 2016, Canada [35] | -- | -- | -- | -- | ✓ | -- | 1 | -- |
| Cancer Care Ontario, 2015, Canada [36] | ✓ | ✓ | ✓ | -- | ✓ | ✓ | 5 | -- |
| American College of Physicians, 2016, United States [37] | -- | -- | -- | -- | ✓ | -- | 1 | -- |
| American Psychiatric Association, 2010, United States [38] | ✓ | ✓ | ✓ | -- | ✓ | ✓ | 5 | ✓ |
| Kaiser Permanente Care Management Institute, 2012 United States [39] | -- | -- | -- | -- | ✓ | ✓ | 2 | ✓ |
| Institute for Clinical Systems Improvement, 2016, United States [40] | ✓ | ✓ | -- | -- | ✓ | ✓ | 4 | ✓ |
| US Preventive Services Task Force, 2016, United States [41] | -- | -- | -- | -- | ✓ | ✓ | 2 | ✓ |
| Scottish Intercollegiate Guidelines Network, 2012, Scotland [42] | -- | ✓ | ✓ | -- | ✓ | ✓ | 4 | -- |
| National Institute for Health and Care Excellence, 2011, England [43] | -- | ✓ | -- | ✓ | ✓ | ✓ | 4 | ✓ |
| National Institute for Health and Care Excellence, 2016, England [44] | ✓ | ✓ | ✓ | ✓ | ✓ | ✓ | 6 | -- |
| Depression total | 7 (38.9) | 12 (66.7) | 9 (50.0) | 5 (27.8) | 15 (83.3) | 15 (83.3) | -- | 9 (50.0) |
| **CARDIAC REHABILITATION** | | | | | | | | |
| American Heart Association: Prevention of Cardiovascular Disease in Women, 2011, United States [45] | -- | ✓ | -- | -- | ✓ | ✓ | 3 | ✓ |
| Heart Failure Society of America, 2017 United States [46] | -- | ✓ | -- | -- | ✓ | -- | 2 | -- |
| National Institute for Health and Care Excellence, 2010, England [47] | -- | ✓ | -- | ✓ | ✓ | ✓ | 4 | ✓ |

*(Continued)*

**Table 2.** (Continued)

| Guideline (year, country) | Patient-centred care domains (n,%) | | | | | | Total PCC domains (n) | Women's health |
|---|---|---|---|---|---|---|---|---|
| | Fostering relationship | Exchanging information | Addressing emotions | Managing uncertainty | Making decisions | Enabling self-management | | |
| Scottish Intercollegiate Guidelines Network, 2016, Scotland [48] | ✓ | ✓ | ✓ | ✓ | ✓ | ✓ | 6 | -- |
| Scottish Intercollegiate Guidelines Network, 2017, Scotland [49] | ✓ | ✓ | ✓ | -- | ✓ | ✓ | 5 | ✓ |
| European Society of Cardiology, 2013, Europe [50] | -- | ✓ | ✓ | -- | -- | -- | 2 | ✓ |
| European Society of Cardiology, 2011, Europe [51] | -- | ✓ | -- | -- | -- | -- | 1 | -- |
| International Council of Cardiovascular Prevention and Rehabilitation, 2016, International [52,53] | -- | -- | ✓ | -- | -- | ✓ | 2 | -- |
| American Heart Association, 2011, United States [54] | -- | ✓ | ✓ | -- | -- | ✓ | 3 | ✓ |
| Cardiac rehabilitation total | 2 (22.2) | 8 (88.9) | 5 (55.6) | 2 (22.2) | 5 (55.6) | 6 (66.7) | -- | 5 (55.6) |
| TOTAL | 9 (33.3) | 20 (74.1) | 14 (51.9) | 7 (25.9) | 20 (74.1) | 21 (77.8) | -- | 14 (51.9) |

encouraged to discuss potential depression or feelings of anxiety with cardiac patients [48–50]. Two (22.2%) guidelines discussed managing uncertainty with respect to living with a cardiac disease and end-of-life [47,48]. Among 5 (55.6%) guidelines that discussed making decisions, most noted that engaging patients in shared decision-making fosters medication adherence [45–49]. The 6 (66.7%) guidelines that mentioned enabling patient self-management outlined that clinicians should provide patients with individualized management that fits their lifestyle, and emphasize exercise programs, smoking cessation, and healthy lifestyle changes.

## Women's health

Of 27 included guidelines, 14 (51.9%) mentioned women's health concepts. Many acknowledged that social determinants of health disproportionately affect women (e.g. economic stability, social support, education), influencing their ability to access high quality care and comply with treatment. However, guidelines provided no frameworks or models of women's health or PCCW, and limited to no guidance on how to support women impacted by these factors, engage women or deliver care to women, or consider life circumstances or preferences specific to women.

**Depression.** Among 18 guidelines, 9 (50.0%) mentioned women's health concepts. Of 3 that focused on peri-/post-natal depression, 2 [28,31] addressed women's health concepts. Key issues of concern for women with depression were higher risk for negative pregnancy outcomes, and the implications of treatment for depression while pregnant and breastfeeding [27,32,38,40,41].

**Cardiac rehabilitation.** Among 9 guidelines, 5 (55.5%) mentioned women's health concepts. Of the two guidelines aimed specifically at women, 1 addressed women's health concepts and concluded that women have different life stresses and responsibilities than men, which may impact treatment adherence [45]. Other guidelines identified that women are generally underrepresented in cardiac research; in particular, women from racial minority groups, leading to programs that may not be specific to women's cardiac needs [45,49,50].

## Guideline quality and PCCW

S3 Table summarizes formal processes or systems used to develop included guidelines. This was not reported for 5 guidelines. Among the remaining 22 guidelines, 14 used their own organization's guideline development manual, 9 used Grading of Recommendations, Assessment, Development and Evaluations, 4 used AGREE, and 3 used the Institute of Medicine guideline development principles. Explicit reporting of use of a guideline development process or system, or process or system used did not appear to be related to inclusion of PCC or women's health concepts in guidelines. Table 3 summarizes AGREE II appraisal of included guidelines. None were recommended; 18 (66.7%) were recommended with modification and 9 (33.3%) were not recommended. For depression guidelines, 14 (77.8%) were recommended with modification and 4 (22.2%) were not recommended. For cardiac rehabilitation guidelines, 4 (44.4%) were recommended with modification and 5 (55.6%) were not recommended.

Scaled domain percentage scores varied widely across guidelines: scope and purpose (52.8% to 100.0%), stakeholder involvement (33.3% to 94.4%), rigor of development (10.4% to 90.6%), clarity of presentation (52.8% to 100.0%), applicability (4.1% to 72.9%) and editorial independence (0.0% to 100.0%). This was also true within and across conditions.

In general, scope and purpose, and clarity of presentation were well-addressed by most guidelines, and applicability scored lower for most guidelines. The stakeholder involvement domain, which reflects the extent to which guidelines were based on the values and preferences of stakeholders including patients, was generally not well-addressed in many guidelines overall (median 61.1%, range 33.3% to 94.4%) and within conditions.

Among 9 (33.3%) of 27 guidelines that scored 70.0% or greater for stakeholder involvement, all were recommended with modifications. Among those 9 guidelines, 7 (77.8%) addressed four or more PCC domains and 7 (77.8%) addressed women's health. For 18 guidelines that scored below 70.0% on stakeholder involvement, 9 (50.0) were not recommended and 9 (50.0%) were recommended with modifications. Among those 18 guidelines, 5 (27.8%) addressed four or more PCC domains and 8 (44.4%) addressed women's health. While not definitive, it appears that stakeholder engagement may increase the likelihood that guidelines address PCCW.

## Discussion

Among 27 guidelines on the conditions of interest published from 2010 to 2017, all mentioned at least one PCC domain and 14 (51.9%) mentioned some aspect of women's health, but none provided comprehensive, detailed or practical information that would help patients and clinicians achieve PCCW. These findings were consistent across guidelines by condition and country. This was also true even when guidelines were specifically aimed at women. Reported formal processes or systems for developing guidelines did not appear to be linked with inclusion of content on PCC or women's health. Based on quality appraisal, guidelines were either not recommended or recommended with modifications. In particular, the stakeholder involvement AGREE II domain was least addressed, but guidelines that scored higher for stakeholder involvement also appeared to better address PCCW. Overall, this research shows that guidelines could be more implementable if they considered PCC and gender.

In general, these findings are concordant with prior research demonstrating variable quality of guidelines [12, 15–17], and with research showing that policies from multiple countries failed to provide guidance on strategies to improve health care quality [55,56]. Given that no prior research has examined guidelines for instructions or support pertaining to PCC or women's health, these findings are novel, and identify opportunities by which to improve guidelines and better support PCCW. Doing so may foster consideration and incorporation of strategies

**Table 3. Quality of included guidelines appraised with AGREE II [17].**

| Guideline (year, country) | Domain score (%) | | | | | | Overall score | Recommendation for use |
|---|---|---|---|---|---|---|---|---|
| | Scope and purpose | Stakeholder involvement | Rigour of development | Clarity of presentation | Applicability | Editorial independence | | |
| **DEPRESSION** | | | | | | | | |
| Canadian Task Force on Preventive Health Care, 2013, Canada [27] | 88.9 | 55.7 | 54.7 | 77.8 | 45.8 | 83.3 | 67.6 | With modifications |
| BC Reproductive Mental Health Program & Perinatal Services BC, 2014, Canada [28] | 77.8 | 52.8 | 43.8 | 88.9 | 29.2 | 12.5 | 50.8 | No |
| Toward Optimized Practice, 2015, Canada [29] | 94.4 | 38.9 | 14.6 | 91.7 | 29.2 | 0.0 | 53.8 | No |
| BC Guidelines, 2013, Canada [30] | 94.4 | 33.3 | 10.4 | 100.0 | 22.9 | 0.0 | 52.2 | No |
| The Centre of Perinatal Excellence, 2017, Australia [31] | 97.2 | 86.1 | 39.6 | 94.4 | 72.9 | 91.7 | 80.3 | With modifications |
| Canadian Partnership Against Cancer and the Canadian Association of Psychosocial Oncology, 2015, Canada [32] | 100.0 | 69.4 | 76.0 | 97.2 | 62.5 | 100.0 | 84.2 | With modifications |
| Royal Australian and New Zealand College of Psychiatrists, 2015, Australia & New Zealand [33] | 97.2 | 94.4 | 56.3 | 72.2 | 37.5 | 75.0 | 72.1 | With modifications |
| Registered Nurse's Association of Ontario, 2016, Canada [34] | 100.0 | 66.7 | 84.4 | 97.2 | 72.9 | 79.2 | 83.4 | With modifications |
| Canadian Network for Mood and Anxiety Treatments, 2016, Canada [35] | 83.3 | 38.9 | 38.5 | 52.8 | 4.1 | 83.3 | 50.2 | No |
| Cancer Care Ontario, 2015, Canada [36] | 97.2 | 61.1 | 90.6 | 72.2 | 47.9 | 70.8 | 73.3 | With modifications |
| American College of Physicians, 2016, United States [37] | 88.9 | 38.9 | 77.1 | 80.6 | 22.9 | 75.0 | 63.9 | With modifications |
| American Psychiatric Association, 2010, United States [38] | 86.1 | 52.8 | 55.2 | 94.4 | 58.3 | 87.5 | 72.4 | With modifications |
| Kaiser Permanente Care Management Institute, 2012 United States [39] | 88.9 | 72.2 | 88.5 | 100.0 | 35.4 | 25.0 | 68.3 | With modifications |
| Institute for Clinical Systems Improvement, 2016, United States [40] | 100.0 | 88.9 | 80.2 | 97.2 | 66.7 | 100.0 | 88.8 | With modifications |
| US Preventive Services Task Force, 2016, United States [41] | 88.9 | 80.6 | 79.2 | 100.0 | 52.1 | 100.0 | 83.4 | With modifications |
| Scottish Intercollegiate Guidelines Network, 2012, Scotland [42] | 100.0 | 61.1 | 78.1 | 100.0 | 72.9 | 25.0 | 72.9 | With modifications |
| National Institute for Health and Care Excellence, 2011, England [43] | 100.0 | 91.7 | 85.4 | 72.2 | 58.3 | 66.7 | 79.1 | With modifications |
| National Institute for Health and Care Excellence, 2016, England [44] | 100.0 | 91.7 | 76.0 | 75.0 | 72.9 | 75.0 | 81.8 | With modifications |
| **CARDIAC REHABILITATION** | | | | | | | | |
| American Heart Association: Prevention of Cardiovascular Disease in Women, 2011, United States [45] | 52.8 | 33.3 | 62.5 | 72.2 | 33.3 | 66.7 | 53.5 | No |

*(Continued)*

**Table 3.** (Continued)

| Guideline (year, country) | Domain score (%) | | | | | | Overall score | Recommendation for use |
|---|---|---|---|---|---|---|---|---|
| | Scope and purpose | Stakeholder involvement | Rigour of development | Clarity of presentation | Applicability | Editorial independence | | |
| Heart Failure Society of America, 2017 United States [46] | 77.8 | 50.0 | 65.6 | 72.2 | 29.2 | 70.8 | 60.9 | With modifications |
| National Institute for Health and Care Excellence, 2010, United Kingdom [47] | 83.3 | 61.1 | 41.7 | 55.6 | 33.3 | 25.0 | 50.0 | No |
| Scottish Intercollegiate Guidelines Network, 2016, Scotland [48] | 100.0 | 83.3 | 90.6 | 97.2 | 50.0 | 41.7 | 77.1 | With modifications |
| Scottish Intercollegiate Guidelines Network, 2017, Scotland [49] | 94.4 | 83.3 | 58.3 | 94.4 | 62.5 | 100.0 | 82.2 | With modifications |
| European Society of Cardiology, 2013, Europe [50] | 77.8 | 55.6 | 26.0 | 80.6 | 29.2 | 25.0 | 49.0 | No |
| European Society of Cardiology, 2011, Europe [51] | 66.7 | 33.3 | 41.7 | 94.4 | 8.3 | 79.2 | 53.9 | No |
| International Council of Cardiovascular Prevention and Rehabilitation, 2016, International [52,53] | 100.0 | 61.1 | 56.3 | 66.7 | 37.5 | 83.3 | 70.25 | With modifications |
| American Heart Association, 2011, United Stat [54] | 61.1 | 44.4 | 60.4 | 97.2 | 2.1 | 91.7 | 59.49 | No |

to support PCCW in guidelines, potentially improving guideline use, and ultimately the health and well-being of women.

One way to do so is to more thoroughly address PCC by considering a PCC framework when generating guidelines. While the McCormack et al. framework may not necessarily be the gold standard,[7] its use revealed that guidelines could be enhanced with information that supports fostering a healing relationship, responding to emotions, or managing uncertainty. To better address PCC, guideline developers could become informed by reviewing PCC literature and models [1–7]. Alternatively, guideline developers could consult with or involve an academic expert in PCC on guideline-writing panels.

Another approach is to identify and incorporate patient perspectives in guidelines. Guidelines informed by patient needs, values and preferences are more likely to be used because they help patients and providers discuss and agree upon the goals of treatment [57–60]. For example, patients who reviewed sickle cell disease guidelines that were informed by preferences gathered from 107 patients said they intended to use the guidelines [61]. Guideline-prompted elicitation of child and caregiver preferences by clinicians resulted in higher asthma medication adherence among patients one month after guideline implementation [62]. Resources are available to help guideline developers understand how to identify and incorporate patient preferences in guidelines by involving patients on guideline-writing panels, interviewing or surveying patients, or reviewing literature on patient preferences pertaining to given guideline topics [63,64].

Yet another approach to enhance guidelines so that they support PCCW is to include guideline implementation tools, defined as information included in or with guidelines that help end-users consider, tailor and apply the recommendations [65]. Few guidelines published before 2010 included implementation tools and they were largely guideline summaries for clinicians [65,66]. Following the issue of criteria and considerations for generating implementation tools [67,68], a higher proportion of recently-developed guidelines included implementation tools of a variety of types for both patients and clinicians [69]. In particular,

implementation tools for patients can inform and/or activate patients and can include information about conditions, lifestyle advice, psychological strategies, and strategies for communicating with clinicians [70].

Strengths of this study include the use of rigorous methods such as a comprehensive search of multiple databases employing a broad search strategy to avoid missing relevant guidelines, independent screening and data extraction, compliance with standards for the reporting of reviews [21], and use of an established PCC framework upon which to map guideline content [7]. Several factors may limit the interpretation and application of the findings. Despite having conducted a comprehensive search of multiple databases we may not have identified all relevant guidelines, plus our search was restricted to English-language guidelines. Furthermore, our review included guidelines on two clinical topics only, thus it is not known if the findings are transferrable to guidelines on other clinical topics. The PCC framework we employed is not necessarily a gold standard, and what constitutes PCC may differ by condition.

Despite a global emphasis on PCC [1–8, 25,26, 71]; and recommendations issued in Canada [13], the United States [72,73], and internationally by the World Health Organization [9,12,21] for greater consideration of women's health; and guideline development standards specifying that guidelines address target user needs and preferences [14], organizations that develop guidelines may need to establish policies or requirements that guidelines address a gendered approach to PCC. Future research should examine whether the findings revealed by our research also pertain to guidelines on other conditions. However, in order for guidelines to address PCCW, research must be available on what constitutes PCCW for different conditions. We conducted a theoretical, rapid review of primary studies on PCCW in the conditions addressed by guidelines included in this study, and identified a paucity of research [74]. Thus, future primary research is needed to identify and compare PCCW across conditions to more thoroughly identify elements that may be broadly relevant, and the characteristics of elements that must be tailored to specific conditions or health care issues.

## Supporting information

**S1 Checklist. PRISMA diagram.**
(DOCX)

**S1 Table. Search strategies.**
(DOCX)

**S2 Table. Patient-centred care domains.**
(DOCX)

**S3 Table. Data extracted from included studies.**
(DOCX)

## Acknowledgments

The following research associates assisted with data collection and manuscript preparation: Mahrukh Zahid, Jessica Ramlakhan, Bryanna Nyhof and Dalia Kagramanov.

## Author Contributions

**Conceptualization:** Anna R. Gagliardi, Courtney Green, Sheila Dunn, Sherry L. Grace, Nazilla Khanlou, Donna E. Stewart.

**Data curation:** Anna R. Gagliardi.

**Formal analysis:** Anna R. Gagliardi.

**Funding acquisition:** Anna R. Gagliardi.

**Investigation:** Anna R. Gagliardi.

**Methodology:** Anna R. Gagliardi, Courtney Green, Sheila Dunn, Sherry L. Grace, Nazilla Khanlou, Donna E. Stewart.

**Project administration:** Anna R. Gagliardi.

**Resources:** Anna R. Gagliardi.

**Supervision:** Anna R. Gagliardi.

**Validation:** Anna R. Gagliardi, Courtney Green, Sheila Dunn, Sherry L. Grace, Nazilla Khanlou, Donna E. Stewart.

**Visualization:** Anna R. Gagliardi, Courtney Green, Sheila Dunn, Sherry L. Grace, Nazilla Khanlou, Donna E. Stewart.

**Writing – original draft:** Anna R. Gagliardi.

**Writing – review & editing:** Anna R. Gagliardi, Courtney Green, Sheila Dunn, Sherry L. Grace, Nazilla Khanlou, Donna E. Stewart.

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
