## [Decision Letter · Decision Letter 0]

7 Oct 2019

PONE-D-19-16686

How do and could clinical guidelines support patient-centred care for women: content analysis of guidelines

PLOS ONE

Dea rAnna R Gagliardi,

Thank you for submitting your manuscript to PLOS ONE. After careful consideration, we feel that it has merit but does not fully meet PLOS ONE’s publication criteria as it currently stands. Therefore, we invite you to submit a revised version of the manuscript that addresses the points raised during the review process.

It would be better if the authors include the GRADE assessment for the overall quality of the guidelines. Scores are very much dependent on the interpretation of the reader. With GRADE, you would need to include on the reasoning for low quality of evidence assessed.

We would appreciate receiving your revised manuscript by Nov 21 2019 11:59PM. To enhance the reproducibility of your results, we recommend that if applicable you deposit your laboratory protocols in protocols.io, where a protocol can be assigned its own identifier (DOI) such that it can be cited independently in the future. For instructions see: http://journals.plos.org/plosone/s/submission-guidelines#loc-laboratory-protocols

We look forward to receiving your revised manuscript.

Kind regards,

Mohd Noor Norhayati, M.B.B.S., M.Comm.Med., Ph.D.

Academic Editor

PLOS ONE

Journal Requirements:

Additional Editor Comments (if provided):

Reviewers' comments:

Reviewer's Responses to Questions

**Comments to the Author**

1. Is the manuscript technically sound, and do the data support the conclusions?

Reviewer #1: Yes

2. Has the statistical analysis been performed appropriately and rigorously? 

Reviewer #1: N/A

3. Have the authors made all data underlying the findings in their manuscript fully available?

Reviewer #1: Yes

4. Is the manuscript presented in an intelligible fashion and written in standard English?

Reviewer #1: Yes

5. Review Comments to the Author

Reviewer #1: A well-done review which raises important issues for guidelines developers and users. I have two questions/suggestions:

1) Why include guidelines that are focused on pregnancy or the peri-partum period (e.g, references 28, 31, 50) when quantifying the proportion of guidelines that address women's health? By definition, they are explicitly addressing women's health, and explicitly for aspects of depression and cardiovascular disease for which there is no possible gender disparity, so they don't really address issues of disparities (and indeed their inclusion leads to an overestimate of the overall proportion addressing women's health). Although the sample size would preclude any formal analysis, I think it would be of great interest to explicitly compare the degree to which these guidelines addressed PCCW and AGREE II standards, versus those that addressed aspects of the conditions affecting both genders (e.g, are guidelines for post-natal depression "better" on domains relevant to PCCW/guideline quality than general guidelines for depression in adults?)

2) There are a number of different approaches to guidelines development, including formal systems such as GRADE, or the approach taken by the USPSTF. Although some of this is captured in the AGREE rigour of development score, it would be helpful to at least identify those which were developed using GRADE (which is meant to be applicable to any guidelines developer) vs some other process.

6. PLOS authors have the option to publish the peer review history of their article (what does this mean?). If published, this will include your full peer review and any attached files.

Reviewer #1: Yes: Evan Myers, MD, MPH

---

## [Author Response · Author response to Decision Letter 0]

12 Oct 2019

EDITOR

It would be better if the authors include the GRADE assessment for the overall quality of the guidelines. Scores are very much dependent on the interpretation of the reader. With GRADE, you would need to include on the reasoning for low quality of evidence assessed.

Author response:

AGREE is used to appraise quality when the unit of analysis is the guideline. GRADE is used to appraise quality of individual studies (unit of analysis) when they are included in guidelines. However, to address this point, please see our thorough response to peer reviewer comment #3. We now report on the process or system used to develop guidelines included in our review (GRADE and others) as an indirect means of commenting on guideline quality, and whether that was associated with inclusion of PCCW content. Also see brief edits in Methods, Data Extraction and Data analysis that explains these updates.

REVIEWER #1

1/ 

A well-done review which raises important issues for guidelines developers and users.

Author Response: Thank you!

2/ 

Why include guidelines that are focused on pregnancy or the peri-partum period (e.g, references 28, 31, 50) when quantifying the proportion of guidelines that address women's health? By definition, they are explicitly addressing women's health, and explicitly for aspects of depression and cardiovascular disease for which there is no possible gender disparity, so they don't really address issues of disparities (and indeed their inclusion leads to an overestimate of the overall proportion addressing women's health). 

Author response:

We now emphasize that while depression was our focus, we included guidelines on depression across the lifespan, thus explaining why some depression guidelines pertained to postnatal depression. Also, we now more explicitly specify evidence of gendered disparities in depression and cardiovascular management. We clarified these issues by changing “We included guidelines on [two] conditions: depression (including post-natal) and cardiovascular care (including cardiac rehabilitation). We chose these topics because they have been associated with known gendered inequities in quality of care…” to:

“We included guidelines on two conditions that affect both men and women across the lifespan: depression (often present in the postnatal period among women) and cardiovascular disease including rehabilitation (now affecting women in middle age). We chose these topics because they have been associated with known gendered inequities in quality of care in Canada and elsewhere: when women report depression, it is more likely to be dismissed as stress compared with men who are more likely to receive treatment; and compared with men, women are less likely to be referred to cardiac rehabilitation [22-24]. These topics were also recommended by….”

Methods, Eligibility Criteria (page 6)

In the original submission, we inadvertently stated that 3 clinical topics were included but had only included 2 (depression including post-natal, cardiovascular disease/treatment/rehab). Thus, we changed “We included guidelines on three conditions…” to “We included guidelines on two conditions…”

S3 Table

In the original submission, we inadvertently uploaded the incorrect version, which included studies on family planning/contraception; those were deleted and we uploaded a revised S3 Table that includes only studies pertaining to depression and cardiovascular disease 

3/

Although the sample size would preclude any formal analysis, I think it would be of great interest to explicitly compare the degree to which these guidelines addressed PCCW and AGREE II standards, versus those that addressed aspects of the conditions affecting both genders (e.g, are guidelines for post-natal depression "better" on domains relevant to PCCW/guideline quality than general guidelines for depression in adults?)

Author response:

Thank you for suggesting this interesting comparison. Please see added content on:

Page 11

Three of 18 depression guidelines focused on peri- or post-natal depression; those guidelines featured content for 2 [28], 4 [42] and 6 [31] PCC domains. Two of 9 cardiovascular disease guidelines featured content for 1 [51] and 3 [45] PCC domains. Thus, guidelines aimed at women did not apparently differ in PCC content from guidelines relevant to both women and men. 

Page 17

Among 18 guidelines, 9 (50.0%) mentioned women’s health concepts. Of those that focused on peri-/post-natal depression, 2 [28,31] of 3 [42] addressed women’s health concepts.

Page 18

Among 9 guidelines, 5 (55.5%) mentioned women’s health concepts. Of the two guidelines aimed specifically at women [45,50], 1 addressed women’s health concepts [45].

Discussion (page 21-22)

Among 27 guidelines on the conditions of interest published from 2010 to 2017, all mentioned at least one PCC domain and 14 (51.9%) mentioned some aspect of women’s health, but none provided comprehensive, detailed or practical information that would help patients and clinicians achieve PCCW. **This was also true even when guidelines were specifically aimed at women.**

Abstract/Results

These findings pertained even to women-specific guidelines.

4/

There are a number of different approaches to guidelines development, including formal systems such as GRADE, or the approach taken by the USPSTF. Although some of this is captured in the AGREE rigour of development score, it would be helpful to at least identify those which were developed using GRADE (which is meant to be applicable to any guidelines developer) vs some other process.

Author response:

Thank you again for suggesting this interesting way to indirectly assess the quality of included guidelines apart from our appraisal with AGREE. We now report the process or system used to develop each guidelines in S3 Table, and summarize the Results in the manuscript as follows: 

Page 18

S3 Table summarizes formal processes or systems used to develop included guidelines. This was not reported for 5 guidelines. Among the remaining 22 guidelines, 14 used their own organization’s guideline development manual, 9 used Grading of Recommendations, Assessment, Development and Evaluations, 4 used AGREE, and 3 used the Institute of Medicine guideline development principles. Explicit reporting of use of a guideline development process or system, or process or system used did not appear to be related to inclusion of PCC or women’s health concepts in guidelines.

Discussion (page 22)

Reported formal processes or systems for developing guidelines did not appear to be linked with inclusion of content on PCC or women’s health.

Abstract/Results

Reported use or type of guideline development process/system did not appear to be linked with PCCW content.

---

## [Editor Report · Decision Letter 1]

16 Oct 2019

How do and could clinical guidelines support patient-centred care for women: content analysis of guidelines

PONE-D-19-16686R1

Dear Dr.Gagliardi,

We are pleased to inform you that your manuscript has been judged scientifically suitable for publication and will be formally accepted for publication once it complies with all outstanding technical requirements.

With kind regards,

Mohd Noor Norhayati, M.B.B.S., M.Comm.Med., Ph.D.

Academic Editor

PLOS ONE
---

## [Editor Report · Acceptance letter]

1 Nov 2019

PONE-D-19-16686R1 

How do and could clinical guidelines support patient-centred care for women: content analysis of guidelines 

Dear Dr. Gagliardi:

I am pleased to inform you that your manuscript has been deemed suitable for publication in PLOS ONE. Congratulations! Your manuscript is now with our production department. 

With kind regards,

on behalf of

Associate Professor Mohd Noor Norhayati 

Academic Editor

PLOS ONE